# Variability in sodium content of takeaway foods: Implications for public health and nutrition policy

Alexandra Irina Mavrochefalos[1,2], Andrew Dodson[3], Gunter G. C. Kuhnle[1,3]*

**1** Department of Food and Nutritional Sciences, University of Reading, Reading, United Kingdom,
**2** Institute for Global Food Security, School of Biological Sciences, Queen's University Belfast, Belfast, United Kingdom, **3** Chemical Analysis Facility, University of Reading, Reading, United Kingdom

* g.g.kuhnle@reading.ac.uk

## Abstract

**Background:** Excessive sodium intake is a major modifiable risk factor for cardiovascular disease, yet accurately assessing dietary sodium remains challenging due to food composition variability and inaccurate menu labeling. While menu labels are intended to guide consumers, discrepancies between reported and actual sodium content could undermine their effectiveness.

**Objective:** To evaluate the accuracy of menu-declared sodium content in takeaway foods by comparing reported values with laboratory measurements.

**Design:** A cross-sectional analysis of 39 takeaway food items from 23 outlets in Reading, UK. Sodium content was measured using Inductively Coupled Plasma – Mass Spectrometry (ICP-MS) and compared to menu-declared values.

**Results:** Sodium content varied widely across food categories. Median sodium levels ranged from 0.1 g/100g (chips from fish & chips shop) to 1.6 g/100g (pizza), with some meals exceeding the 6 g/day recommended intake in a single serving. Curry dishes exhibited the greatest variability (2.3–9.4 g per dish). Significant discrepancies were found between menu-reported and measured sodium levels, with almost 50% of foods exceeding declared values.

**Conclusion:** Take-away foods exhibit substantial sodium variability, and menu labels often fail to accurately reflect actual sodium content. These findings have implications for nutritional epidemiology, where inaccurate sodium estimates may misclassify intake, and for public health, as misleading labels could hinder sodium reduction efforts. It is therefore important that menu labels are not considered definitive but rather general indicators of sodium content and potentially other nutrients.

## Introduction

Accurately assessing dietary intake is one of the most challenging aspects of nutritional epidemiology [1]. Food composition is inherently variable due to differences in raw ingredients, food preparation methods, portion sizes, and processing techniques. Furthermore, even within similar food items, substantial variability exists depending

**Data availability statement:** Anonymised data will be available from https://doi.org/10.17864/1947.001409.

**Funding:** The author(s) received no specific funding for this work.

**Competing interests:** The authors have declared that no competing interests exist.

on the source of the food, whether prepared in a chain restaurant with standardized recipes or in an independent outlet with less stringent controls [2].

Food composition databases, which underpin dietary assessment, are often based on averaged values from specific sources and fail to capture the high degree of variability in real-world settings. This is particularly problematic for nutrients like sodium, which is frequently added during food preparation in variable amounts, often without precise measurement. Content can therefore differ substantially between otherwise similar products. Because sodium is a key modifiable risk factor for hypertension and cardiovascular disease, even small discrepancies between labelled and actual values can have significant implications for both individual health and population-level intake estimates [3].

Elevated sodium consumption is causally linked to increased blood pressure, which in turn raises the risk of stroke, coronary heart disease, and other cardiovascular conditions [4]. Meta-analyses of randomised controlled trials have shown that modest, sustained reductions in sodium intake can lead to significant decreases in systolic and diastolic blood pressure across diverse populations [5]. Estimates by the World Health Organization suggest that excess sodium intake contributes to up to 1.8 million deaths annually, making it one of the leading dietary risk factors for mortality and disease burden globally [6]. These risks are particularly relevant for vulnerable populations, including those with hypertension, kidney disease, or other non-communicable conditions. Therefore, improving the accuracy of sodium content information is essential for effective dietary guidance and policy intervention.

Menu labelling has been proposed as a strategy to enhance consumer awareness and support healthier dietary choices [7], and there have been extensive discussions about their impact [8]. While some jurisdictions have implemented mandatory labelling requirements, there is growing evidence that menu labels often do not reflect actual food composition accurately [9]. These inaccuracies arise due to variability in food preparation, ingredient substitutions, and inherent differences in portion sizes. The reliance on estimated values instead of laboratory analyses contributes to discrepancies between reported and actual sodium levels, limiting the effectiveness of menu labelling as a public health intervention.

In this study, we have therefore investigated the accuracy of menu food labelling, using sodium as indicator as reducing sodium intake is a key target for public health, it is an important modifiable risk factor for cardio-vascular diseases and sodium can be reliably measured in food.

## Methods

### Food collection

All foods (n = 3 per item, see Table 1) were purchased from a random selection of take-away outlets across Reading in the County of Royal Berkshire (England, United Kingdom), to capture the diversity of foods typically purchased locally. Foods were

**Table 1**. **Foods included in the analyses.** All foods were purchased in summer 2022.

| Food group | Outlet | Name |
|---|---|---|
| Burger | Chicky | Chicky PeriPeri burger |
| Burger | KFC | KFC Fillet Burger |
| Burger | McDonalds | McDonalds BigMac |
| Burger | Chicken Hut | PeriPeriChicken Hut Burger |
| Burger (vegan) | KFC | KFC vegan burger |
| Burger (vegan) | McDonalds | McDonalds Mcplant |
| Chicken (battered) | Burger King | Burger King Chicken Nuggets |
| Chicken (battered) | GBK | T.O. Gourmet Burger Chicken Nuggets |
| Chicken (battered, vegan) | Burger King | Burger King Vegan Nuggets |
| Chicken (grilled) | Nando's | Nando's 1/4 Chicken medium |
| Chicken (grilled) | Pizza House | Pizza House PeriPeri Chicken |
| Chicken (sauce-based) | Seasons | Seasons Caribbean Jerk Chicken |
| Chips | Chicky | Chicky PeriPeri fries |
| Chips | KFC | KFC Fries |
| Chips | McDonalds | McDonalds medium fries |
| Chips | Chicken Hut | PeriPeriChicken Hut Fries |
| Chips (Fish & Chips) | Finn's | Finn's Fish and Chips - Chips |
| Curry | House of Flavours | House of Flavours Chicken Tikka |
| Curry | Sen Sushi | Sen Sushi Chicken Kaatsu Curry |
| Curry | Wagamama | Wagamama chicken katsu curry |
| Curry (vegan) | House of Flavours | House of Flavours Chana Masala |
| Curry (vegan) | Wagamama | Wagamama vegan katsu curry |
| Fish | Finn's | Finn's Fish and Chips - Cod |
| Fish (vegan) | Finn's | Finn's Fish and Chips - Vegan Fish |
| Pasta dish | Home cooking | Home cooking Meat Chow Mein |
| Pasta dish | Home cooking | Home cooking Veg Chow Mein |
| Pies | Cairo's | Cairo's Vegan Sausage Roll |
| Pies | Dinky's | Dinky's Sausage Roll |
| Pies | Greggs | Greggs Sausage Roll |
| Pies | Greggs | Greggs Vegan Sausage Roll |
| Pizza | Domino's | Domino's Pepperoni Pizza 9" |
| Pizza | Presto | Presto pepperoni pizza 10" |
| Pizza (vegan) | Domino's | Domino's Vegan Pepperoni Pizza 11.5" |
| Sandwich | German Doner Kebab | German Doner Kebab Chicken |
| Sandwich | Reading Kebab | Reading Kebab & Grill Pizza Chicken Kebab |
| Sandwich | Pierre's | Pierre's Ham Baguette |
| Sandwich | Subway | Subway Ham Sub |
| Sandwich (vegan) | German Doner Kebab | German Doner Kebab Veggie |
| Sandwich (vegan) | Subway | Subway Vegan (TLC) Sub |

purchased during three consecutive weeks in summer 2022. Purchases were made anonymously to avoid potential bias in preparation. Upon receipt, samples were weighed, frozen at -20°C, and freeze-dried. A total of 39 different foods from 23 different outlets were analysed.

## Laboratory analysis

Samples were analysed using Inductively Coupled Plasma - Mass Spectrometry (ICP-MS) following AOAC Official Method 2011.14, a validated approach for precise sodium quantification in food matrices. To ensure accuracy, calibration was performed using sodium standards ranging from 500 to 5000 ppb.

   **Grinding.** Freeze-dried food items were ground in a food processor. The food processor, and all tools used were washed HPLC-grade water to avoid cross-contamination. The foods were stored in heat-sealed polyethylene bags.

**Ashing.** Samples were weighed into 50 mL Pyrex conical flasks that had been washed with HPLC-grade water to avoid contamination. Samples were ashed at 550°C for eight hours in a muffle furnace (Carbolite GERO AAF1100).

**Analysis.** Ashed samples were dissolved in nitric acid (Fisher, 67%-69%, Trace Metal Grade) and diluted with HPLC-grade water analysed by ICP-MS for sodium concentration (Thermo Fisher Scientific iCAPQ). Samples were quantified using a six-point calibration curve (0 ppb, 500 ppb, 1000 ppb, 2000 ppb, 2500 ppb, and 5000 ppb). As a control measure, quality control samples and blanks were interspersed regularly (every 10 samples). Quality control samples were prepared by adding a defined amount of sodium to a food sample. For quality assurance, quality control (QC) samples and blanks were analysed one time for every 10 samples.

## Statistical analysis

Data were analysed using R 4.0 [10]. Graphics were created with ggplot2 [11].

## Results

### Variability in salt content across food categories

The measured salt content of takeaway dishes varied significantly between food categories (Fig 1 and Table 2). (In this study, "takeaway" refers to foods prepared by commercial vendors for immediate consumption off the premises, without table service. This includes both hot and cold items sold by fast food outlets, independent vendors, and chain restaurants, but excludes meals intended for home preparation or consumed on-site in full-service establishments). The median salt content per dish ranged from 0.1 g/100g in chips from fish & chip shops to 1.6 g/100g in pizza with some individual dishes exceeding the recommended daily intake of 6 g in a single serving. Pasta dishes and sandwiches contained the highest median salt levels, with some dishes exceeding 6 g per serving. Curry dishes showed the greatest variation in salt content, ranging from 2.3 to 9.4 g per dish, depending on the ingredients and preparation methods.

Fish and chips meals had relatively lower salt levels compared to other categories, though variation was still observed between different vendors. These differences highlight the wide range of sodium content in takeaway foods, emphasizing the need for careful consumer choices when selecting meals.

### Discrepancies between measured and nutritional information

Nutritional information were only available for 17 different types of foods. A comparison between the declared salt content and laboratory-measured values (Fig 2) revealed significant inconsistencies. While some dishes contained less salt than indicated on menus, almost have others had substantially higher levels than reported. Indeed, in almost half of all food categories, at least some of the samples contained more salt than declared and in several instances all foods sampled exceeded the labelled amount.

## Discussion

This study highlights the substantial variability in salt content across different takeaway food categories and the discrepancies between measured sodium levels and the values provided on restaurant menus. The findings demonstrate that many dishes contain salt levels exceeding dietary recommendations, with some individual meals surpassing the 6 g/day recommendation in a single serving. While expected variations exist due to differences in recipe formulations and preparation methods, the observed inconsistencies between reported and actual salt content raise concerns about the accuracy of nutritional labeling and its implications for dietary assessment and public health.

The variability in salt content across food categories can be attributed to several factors. Recipes within the same category often differ in ingredients, portion sizes, and seasoning practices, contributing to wide-ranging sodium levels. Previous studies have shown that processed and prepared foods contribute significantly to total sodium intake and that their composition can vary widely between products and brands [13,14]. The discrepancies between menu-declared sodium

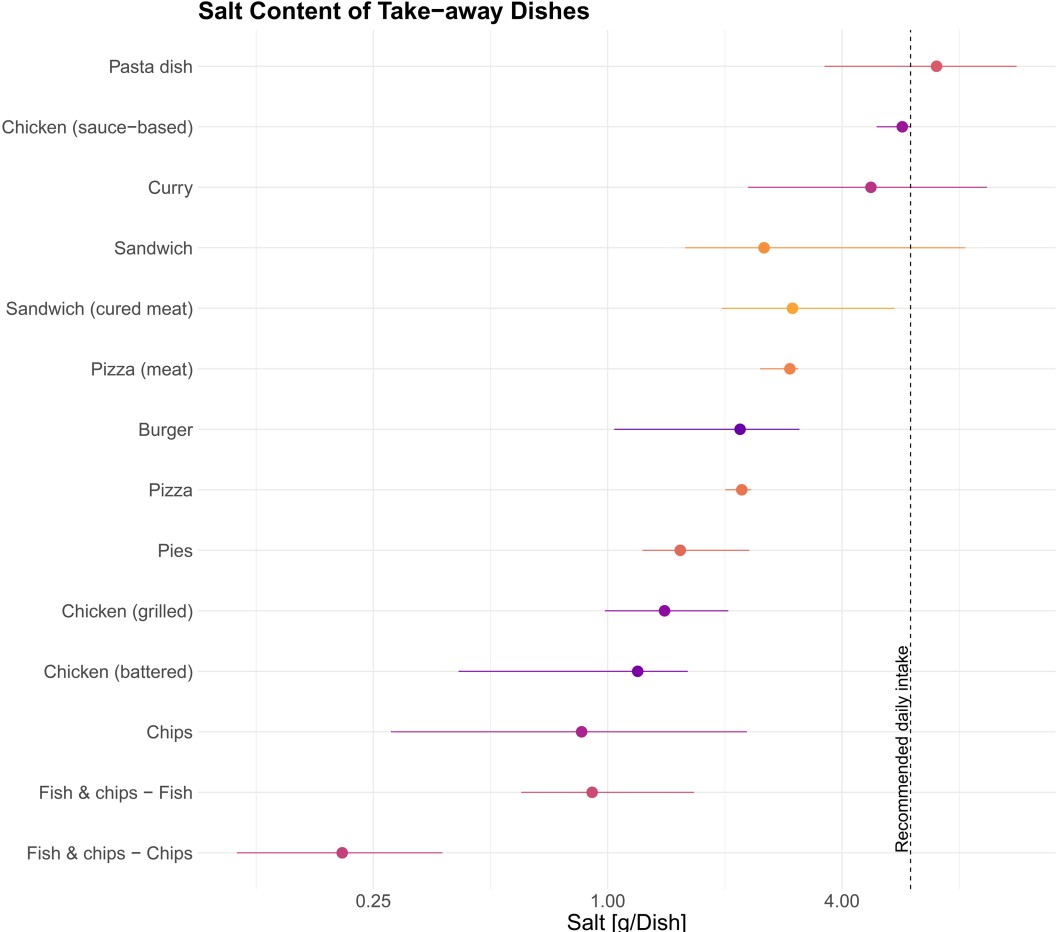

**Fig 1**. **Salt content (mean and range) of takeaway dishes purchased in 2022.** The red line indicates the recommendations for salt intake at the time of purchase [12].

**Table 2**. **Measured sodium content (mean, median, and range) in takeaway food categories.** Sodium values are expressed per 100g and per dish.

| Food | | Sodium in g/100g | | | Sodium content in g/dish | | |
|---|---|---|---|---|---|---|---|
| | n | Mean | Median | Range | Mean | Median | Range |
| Burger | 18 | 1.1 | 1 | 0.6 - 1.6 | 2.2 | 2.2 | 1 - 3.1 |
| Chicken (battered) | 9 | 1 | 1.3 | 0.4 - 1.5 | 1 | 1.2 | 0.4 - 1.6 |
| Chicken (grilled) | 6 | 0.7 | 0.6 | 0.4 - 1.3 | 1.4 | 1.4 | 1 - 2 |
| Chicken (sauce-based) | 3 | 1 | 1 | 0.9 - 1.1 | 5.5 | 5.7 | 4.9 - 6 |
| Chips | 12 | 0.6 | 0.6 | 0.2 - 1.1 | 1 | 0.9 | 0.3 - 2.3 |
| Curry | 15 | 0.8 | 0.6 | 0.5 - 1.8 | 4.9 | 4.7 | 2.3 - 9.4 |
| Fish & chips - Chips | 3 | 0.1 | 0.1 | 0 - 0.1 | 0.2 | 0.2 | 0.1 - 0.4 |
| Fish & chips - Fish | 6 | 0.8 | 0.7 | 0.4 - 1.3 | 1 | 0.9 | 0.6 - 1.7 |
| Pasta dish | 6 | 1.1 | 1.1 | 0.7 - 1.5 | 7.2 | 7 | 3.6 - 11.2 |
| Pies | 12 | 1.5 | 1.5 | 1.1 - 1.8 | 1.6 | 1.5 | 1.2 - 2.3 |
| Pizza | 3 | 1.2 | 1.2 | 1.2 - 1.3 | 2.2 | 2.2 | 2 - 2.3 |
| Pizza (meat) | 6 | 1.6 | 1.6 | 1.3 - 1.7 | 2.8 | 2.9 | 2.5 - 3.1 |
| Sandwich | 12 | 0.9 | 0.9 | 0.5 - 1.4 | 3.4 | 2.5 | 1.6 - 8.3 |
| Sandwich (cured meat) | 6 | 1.5 | 1.4 | 1.3 - 1.9 | 3.3 | 3 | 2 - 5.5 |

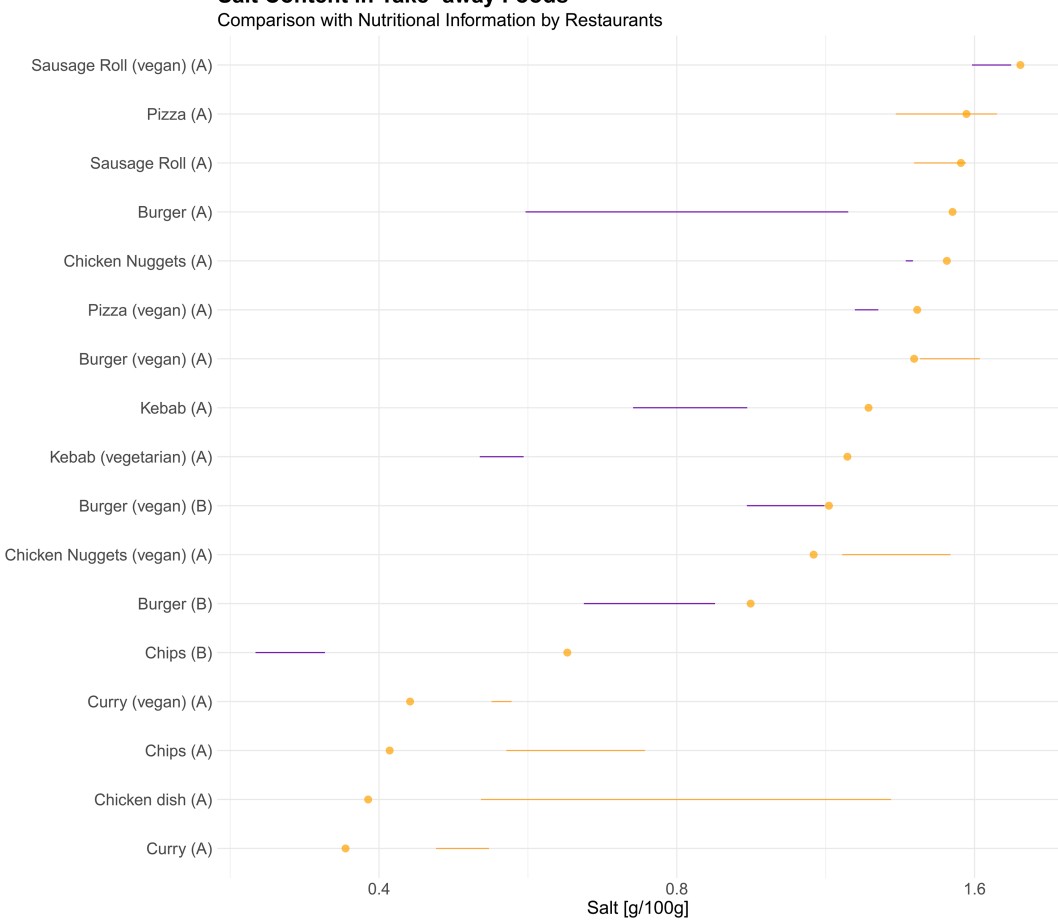

**Fig 2**. **Comparison between measured food content and menu label in various food items.** The diamond indicates sodium content according to the nutrition information provided.

content and laboratory measurements are a key finding of this study. In contrast to data from the US [15], we found considerable differences between nutritional information provided by restaurants and actual salt content. This is likely a consequence of the limitation of using static food composition databases and ignoring the dynamic nature of food composition.

One of the strengths of this study is its reliance on laboratory analysis rather than calculated estimates, ensuring a more accurate assessment of the actual sodium content in foods. Additionally, the study captures a diverse range of food categories and provides insights into both per-100g and per-dish salt content, which are essential for assessing total intake. However, there are also some limitations: the sample size for certain food categories was small, limiting the ability to generalize findings across the entire takeaway sector. Furthermore, as samples were collected from selected locations, regional and vendor-specific differences in salt content may not be fully accounted for.

The implications of these findings extend beyond consumer awareness to nutritional epidemiology and public health policy [12]. From a nutrition research perspective, these results reinforce the importance of using 24h urine samples and not dietary data to estimate sodium intake. A reliance on food content data — even of foods often considered to be standardised — is likely going to affect outcomes as we have shown previously for other bioactive compounds [16].

This study focuses on the UK, where salt-reduction initiatives and product-specific benchmarks have been in place for many years [12]. Although our sample is small and not designed to be representative of all UK foods, it illustrates the degree of compositional variability that is likely to occur more widely, given that production processes are broadly similar across settings. A larger, country-specific investigation would be required to assess compliance with international WHO benchmarks [17].

From a public health standpoint, the discrepancies in salt labelling raise concerns about the reliability of menu-based nutrition guidance. Like sodium, sugar has many technological roles in food and therefore sugar content can depend on a range of different factors [18]. Consumers relying on menu labels to make healthier choices may unknowingly exceed recommended sodium intake due to inaccurate or outdated information. This is particularly relevant for individuals advised to follow low-sodium diets, as misleading menu labels may compromise dietary adherence. Given that high sodium intake is a major risk factor for hypertension and cardiovascular disease, improving the accuracy of menu labels could contribute to sodium reduction initiatives and more effective public health interventions [5].

## Conclusion

This study underscores the complexity of real-world sodium intake and the limitations of current menu labelling practices. The findings contribute to a better understanding of the *dynamic foodome*, where food composition is subject to multiple influences, including preparation methods, ingredient variability, and supply chain changes. As such, menu labels should not be considered definitive but rather as general indicators of sodium content and other nutrients.

## Acknowledgments

GGCK would like to thank CVA and JH for their support analysing data and drafting this manuscript.

## Author contributions

**Conceptualization:** Alexandra Irina Mavrochefalos, Gunter G. C. Kuhnle.

**Data curation:** Gunter G. C. Kuhnle.

**Formal analysis:** Alexandra Irina Mavrochefalos, Gunter G. C. Kuhnle.

**Investigation:** Alexandra Irina Mavrochefalos.

**Methodology:** Alexandra Irina Mavrochefalos, Gunter G. C. Kuhnle, Andrew Dodson.

**Visualization:** Gunter G. C. Kuhnle.

**Writing – original draft:** Gunter G. C. Kuhnle.

**Writing – review & editing:** Gunter G. C. Kuhnle.

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
