## [Decision Letter · Decision Letter 0]

18 Jul 2025

PONE-D-25-12820Variability in Sodium Content of Takeaway Foods: Implications for Public Health and Nutrition PolicyPLOS ONE

Dear Dr. Kuhnle,

Thank you for submitting your manuscript to PLOS ONE. After careful consideration, we feel that it has merit but does not fully meet PLOS ONE’s publication criteria as it currently stands. Therefore, we invite you to submit a revised version of the manuscript that addresses the points raised during the review process.

We look forward to receiving your revised manuscript.

Kind regards,

Eric Nyarko, BSc, MPhil, PhD, MPH

Academic Editor

PLOS ONE

Journal Requirements:

2. We note you have included a table to which you do not refer in the text of your manuscript. Please ensure that you refer to Table 1 in your text; if accepted, production will need this reference to link the reader to the Table.

Reviewers' comments:

Reviewer's Responses to Questions

**Comments to the Author**

1. Is the manuscript technically sound, and do the data support the conclusions?

Reviewer #1: Yes

Reviewer #2: No

2. Has the statistical analysis been performed appropriately and rigorously?

Reviewer #1: Yes

Reviewer #2: Yes

3. Have the authors made all data underlying the findings in their manuscript fully available?

Reviewer #1: Yes

Reviewer #2: No

4. Is the manuscript presented in an intelligible fashion and written in standard English?

Reviewer #1: Yes

Reviewer #2: Yes

5. Review Comments to the Author

Reviewer #1: Dear Authors,

Thank you for the opportunity to review your work. This is an important study with implications for public health. The study is well executed and straightforward. I have the following comments for your consideration:

Please clarify why the failure of food databases to capture the high degree of variablity in nutrient content is particularly important for sodium (page 1, last sentence).

Please discuss the adverse health effects of an excessive sodium intake in greater detail.

Figure 2 is interesting but I believe a Bland-Altman Difference Plot (e.g., this: https://www.techtips.surveydesign.com.au/post/a-bland-altman-difference-plot-in-stata) would capture your findings in a more adequate way.

Please double-check references for correctness.

Reviewer #2: The manuscript aims to evaluate the sodium content in some category products – labelled as “takeaway products” – sold in Reading, UK and compare it with the sodium/salt content present on the food labelling.

I am sincerely missing which is the first aim of this manuscript as, from one side authors seem to evidence the difference in the labelled sodium content vs the analyzed, but then they are also claiming – starting from the title – a public health and nutrition policy aim.

First of all, the “takeaway” product category has no scientific meaning or definition. So, each product which is prepared somewhere for selling and kept away from that place can be takeaway. Of course in takeaway there can be present products not packaged, and so authors cannot compare the sodium content. Then, in table 1 there are some “cherry picked” products with different recipes, which can differ from each other also in the sale from different markets.

Then, I cannot understand which is the public health and nutrition policy. Authors are not referring to documents which set benchmarks for the “takeaway” products, i.e. https://www.who.int/publications/i/item/9789240092013, and are not comparing such values with the reference. More, authors are not suggesting what it can be done to reduce the salt and reformulate the product (if it can be reformulated).

6. PLOS authors have the option to publish the peer review history of their article (what does this mean?). If published, this will include your full peer review and any attached files.

Reviewer #1: **Yes: **Maximilian Andreas Storz

Reviewer #2: No

---

## [Decision Letter · Decision Letter 1]

2 Aug 2025

PONE-D-25-12820R1Variability in Sodium Content of Takeaway Foods: Implications for Public Health and Nutrition PolicyPLOS ONE

Dear Dr. Kuhnle,

Thank you for submitting your manuscript to PLOS ONE. After careful consideration, we feel that it has merit but does not fully meet PLOS ONE’s publication criteria as it currently stands. Therefore, we invite you to submit a revised version of the manuscript that addresses the points raised during the review process.

We look forward to receiving your revised manuscript.

Kind regards,

Eric Nyarko, BSc, MPhil, PhD, MPH

Academic Editor

PLOS ONE

Journal Requirements:

Reviewers' comments:

Reviewer's Responses to Questions

**Comments to the Author**

1. If the authors have adequately addressed your comments raised in a previous round of review and you feel that this manuscript is now acceptable for publication, you may indicate that here to bypass the “Comments to the Author” section, enter your conflict of interest statement in the “Confidential to Editor” section, and submit your "Accept" recommendation.

Reviewer #1: All comments have been addressed

Reviewer #2: (No Response)

2. Is the manuscript technically sound, and do the data support the conclusions?

Reviewer #1: Yes

Reviewer #2: No

3. Has the statistical analysis been performed appropriately and rigorously?

Reviewer #1: Yes

Reviewer #2: No

4. Have the authors made all data underlying the findings in their manuscript fully available?

Reviewer #1: Yes

Reviewer #2: Yes

5. Is the manuscript presented in an intelligible fashion and written in standard English?

Reviewer #1: Yes

Reviewer #2: Yes

6. Review Comments to the Author

Reviewer #1: Dear Authors,

Thank you for your detailed point-by-point response. I have no further comments.

Best wishes

Reviewer #2: The revision of the authors substantially not considered any of the aspect rose up by the referee, including the comparison with WHO benchmark neither an explication of the selection of the products, which of course cannot be a reference for UK product, but such product have no real selection method.

7. PLOS authors have the option to publish the peer review history of their article (what does this mean?). If published, this will include your full peer review and any attached files.

Reviewer #1: **Yes: **Maximilian Andreas Storz

Reviewer #2: No

---

## [Author Response · Author response to Decision Letter 2]

6 Nov 2025

We appreciate the reviewer’s suggestion regarding WHO benchmarks. While these global reference values are important, compliance with UK salt-reduction targets was more relevant for this study, as they are the operational standards guiding product reformulation in the UK. To avoid confusion between policy levels, WHO val ues were mentioned only in the Discussion.

The product selection process has now been clarified (Methods): “All foods (n=3 per item, see Table 1) were purchased from a random selection of take-away outlets across Reading in the County of Royal Berkshire (England, United Kingdom), to capture the diversity of foods typically purchased locally.”

---

## [Decision Letter · Decision Letter 2]

1 Dec 2025

PONE-D-25-12820R2Variability in Sodium Content of Takeaway Foods: Implications for Public Health and Nutrition PolicyPLOS ONE

Dear Dr. Kuhnle,

Thank you for submitting your manuscript to PLOS ONE. After careful consideration, we feel that it has merit but does not fully meet PLOS ONE’s publication criteria as it currently stands. Therefore, we invite you to submit a revised version of the manuscript that addresses the points raised during the review process.

We look forward to receiving your revised manuscript.

Kind regards,

António Raposo

Academic Editor

PLOS ONE

Journal Requirements:

Reviewers' comments:

Reviewer's Responses to Questions

**Comments to the Author**

1. If the authors have adequately addressed your comments raised in a previous round of review and you feel that this manuscript is now acceptable for publication, you may indicate that here to bypass the “Comments to the Author” section, enter your conflict of interest statement in the “Confidential to Editor” section, and submit your "Accept" recommendation.

Reviewer #2: (No Response)

Reviewer #3: All comments have been addressed

Reviewer #4: (No Response)

2. Is the manuscript technically sound, and do the data support the conclusions?

Reviewer #2: No

Reviewer #3: Yes

Reviewer #4: Yes

3. Has the statistical analysis been performed appropriately and rigorously?

Reviewer #2: No

Reviewer #3: Yes

Reviewer #4: Yes

4. Have the authors made all data underlying the findings in their manuscript fully available?

Reviewer #2: Yes

Reviewer #3: Yes

Reviewer #4: Yes

5. Is the manuscript presented in an intelligible fashion and written in standard English?

Reviewer #2: Yes

Reviewer #3: Yes

Reviewer #4: Yes

6. Review Comments to the Author

Reviewer #2: I am concerned that all the questions rose up in the past revision steps have not been taken into consideration.

Concerning the first comment, I cannot see any WHO benchmark related mention in conclusion, and also references in the text are not available, as I can only see a series of "?" instead of numbero of reference.

Concerning the second comment "a random selection of the products" cannot be a proof of "to capture the diversity of foods typically purchased locally. " In fact, being random chosen, it cannot be covered the differences in terms of salt neither in terms of diffusion of the products neither in terms of consumption.

Reviewer #3: (No Response)

Reviewer #4: I found your work to be highly relevant, well-structured, and significantly important in addressing the challenges associated with dietary sodium intake and menu labeling accuracy in takeaway foods.

while I believe the manuscript is already of high quality, I would make the below minor points;

1) there is slight discrepancy in your recommendation in Abstract and the main document- ''Regular validation of menu labels and improved consumer guidance are needed'' this did not appear in the main document but in manuscript.

2) in your introductory part line 21-23(Estimates by the World Health 21

Organization suggest that excess sodium intake contributes to up to 1.8 million deaths 22

annually, making it one of the leading dietary risk factors for mortality and disease 23

burden globally) it is good that you have glimpsed from global perspective and it would be good if relate and connect to the global salt reduction target

7. PLOS authors have the option to publish the peer review history of their article (what does this mean?). If published, this will include your full peer review and any attached files.

Reviewer #2: No

Reviewer #3: No

Reviewer #4: No

---

## [Author Response · Author response to Decision Letter 3]

2 Dec 2025

R2:

Concerning the first comment, I cannot see any WHO benchmark related mention in conclusion, and also references in the text are not available. Concerning the second comment “a random selection of theproducts” cannot be a proof of “to capture the diversity of foods typically purchased locally. ” In fact, being random chosen, it cannot be covered the differences in terms of salt neither in terms of diffusion

Response: We agree that a simple random purchase cannot guarantee representativeness of local dietary patterns. Our intention was narrower: to avoid subjective selection of outlets or products and to reflect what is available for purchase in the study area. This has been made clear throughout the text and we never claim that this sample is representative.

We have added the following paragraph to the discussion:

This study focuses on the UK, where salt reduction initiatives and product-specific benchmarks have been in place for many years [UK Benchmark]. Although our sample is small and not designed to be representative of all UK foods, it illustrates the degree of compositional variability that is likely to occur more widely, given that production processes are broadly similar across settings. A larger, country-specific investigation would be required to assess compliance with international WHO benchmarks [WHO Benchmark].

R4.1

there is slight discrepancy in your recommendation in Abstract and the main document- “Regular validation of menu labels and improved consumer guidance are needed” this did not appear in the main document but in manuscript.

In order to align with the main conclusion of the study — i.e. that content labels on food items should be seen more as indicative and not definitive amount — we have changed the abstract and replaced the last sentence with “It is therefore important that menu labels are not considered definitive but rather general indicators of sodium content and potentially other nutrients.”

R4.2

Thee World Health Organization suggest that excess sodium intake contributes to up to 1.8 million deaths annually, making it one of the leading dietary risk factors for mortality and disease 23 burden globally) it is good that you have glimpsed from global perspective and it would be good if relate and connect to the global salt reduction target

Thank you for this comment, and we appreciate the importance of recognising international benchmarks. However, as the UK has a dedicated salt-reduction campaign with specific benchmarks, we believe it would be potentially misleading to make any claims beyond the UK.

We have added the following paragraph to the discussion:

This study focuses on the UK, where salt reduction initiatives and product-specific benchmarks have been in place for many years [UK Benchmark]. Although our sample is small and not designed to be representative of all UK foods, it illustrates the degree of compositional variability that is likely to occur more widely, given that production processes are broadly similar across settings. A larger, country-specific investigation would be required to assess compliance with international WHO benchmarks [WHO Benchmark].

---

## [Decision Letter · Decision Letter 3]

7 Dec 2025

Variability in Sodium Content of Takeaway Foods: Implications for Public Health and Nutrition Policy

PONE-D-25-12820R3

Dear Dr. Kuhnle,

We’re pleased to inform you that your manuscript has been judged scientifically suitable for publication and will be formally accepted for publication once it meets all outstanding technical requirements.

Kind regards,

António Raposo

Academic Editor

PLOS One

Additional Editor Comments (optional):

Reviewers' comments:

Reviewer's Responses to Questions

**Comments to the Author**

1. If the authors have adequately addressed your comments raised in a previous round of review and you feel that this manuscript is now acceptable for publication, you may indicate that here to bypass the “Comments to the Author” section, enter your conflict of interest statement in the “Confidential to Editor” section, and submit your "Accept" recommendation.

Reviewer #5: All comments have been addressed

2. Is the manuscript technically sound, and do the data support the conclusions?

Reviewer #5: Yes

3. Has the statistical analysis been performed appropriately and rigorously?

Reviewer #5: Yes

4. Have the authors made all data underlying the findings in their manuscript fully available?

Reviewer #5: Yes

5. Is the manuscript presented in an intelligible fashion and written in standard English?

Reviewer #5: Yes

6. Review Comments to the Author

Reviewer #5: (No Response)

7. PLOS authors have the option to publish the peer review history of their article (what does this mean?). If published, this will include your full peer review and any attached files.

Reviewer #5: **Yes: **M. João Reis Lima

---

## [Editor Report · Acceptance letter]

PONE-D-25-12820R3

PLOS One

Dear Dr. Kuhnle,

I'm pleased to inform you that your manuscript has been deemed suitable for publication in PLOS One. Congratulations! Your manuscript is now being handed over to our production team.

Kind regards,

on behalf of

Dr. António Raposo

Academic Editor

PLOS One